# Future trends in incidence and long-term survival of metastatic cancer in the United States

Nicholas L. Hudock [1,2], Kyle Mani[3], Chachrit Khunsriraksakul[1,2,4], Vonn Walter[5], Larissa Nekhlyudov[6], Ming Wang [7], Eric J. Lehrer [8], Maria R. Hudock [9,10], Dajiang J. Liu [4,5], Daniel E. Spratt[11] & Nicholas G. Zaorsky[11✉]

**Background** Previous studies have demonstrated epidemiological trends in individual metastatic cancer subtypes; however, research forecasting long-term incidence trends and projected survivorship of metastatic cancers is lacking. We assess the burden of metastatic cancer to 2040 by (1) characterizing past, current, and forecasted incidence trends, and (2) estimating odds of long-term (5-year) survivorship.

**Methods** This retrospective, serial cross-sectional, population-based study used registry data from the Surveillance, Epidemiology, and End Results (SEER 9) database. Average annual percentage change (AAPC) was calculated to describe cancer incidence trends from 1988 to 2018. Autoregressive integrating moving average (ARIMA) models were used to forecast the distribution of primary metastatic cancer and metastatic cancer to specific sites from 2019 to 2040 and JoinPoint models were fitted to estimate mean projected annual percentage change (APC).

**Results** The average annual percent change (AAPC) in incidence of metastatic cancer decreased by 0.80 per 100,000 individuals (1988–2018) and we forecast an APC decrease by 0.70 per 100,000 individuals (2018–2040). Analyses predict a decrease in metastases to liver (APC = −3.40, 95% CI [−3.50, −3.30]), lung (APC (2019–2030) = −1.90, 95% CI [−2.90, −1.00]); (2030–2040) = −3.70, 95% CI [−4.60, −2.80]), bone (APC = −4.00, 95% CI [−4.30, −3.70]), and brain (APC = −2.30, 95% CI [−2.60, −2.00]). By 2040, patients with metastatic cancer are predicted to have 46.7% greater odds of long-term survivorship, driven by increasing plurality of patients with more indolent forms of metastatic disease.

**Conclusions** By 2040, the distribution of metastatic cancer patients is predicted to shift in predominance from invariably fatal to indolent cancers subtypes. Continued research on metastatic cancers is important to guide health policy and clinical intervention efforts, and direct allocations of healthcare resources.

### Plain Language Summary

Cancer that has spread beyond the area where it originated and into different organs is called metastatic cancer. This study analyzed trends in metastatic cancer incidence, the proportion of those with metastatic cancer surviving 5 years after diagnosis and the locations in the body each cancer had spread to. The incidence of metastatic cancer decreased between 1988 and 2018 and is expected to continue to decrease until 2040. Some of the most common locations cancer spreads to is the lung, liver, brain, and bone. Metastatic cancer incidence to these areas is predicted to decrease. Also, the likelihood of surviving for more than 5 years after diagnosis with metastatic cancer is predicted to increase by 2040. This research should facilitate optimal planning of future healthcare resources and policy.

[1] Department of Radiation Oncology, Penn State Cancer Institute, Hershey, PA, USA. [2] Penn State College of Medicine, Hershey, PA, USA. [3] Albert Einstein School of Medicine, Bronx, NY, USA. [4] Department of Bioinformatics and Genomics, Penn State College of Medicine, Hershey, PA, USA. [5] Department of Public Health Sciences, Penn State College of Medicine, Hershey, PA, USA. [6] Department of Internal Medicine, Harvard Medical School, Boston, MA, USA. [7] Department of Population and Quantitative Health Sciences, Case Western Reserve University School of Medicine, Cleveland, OH, USA. [8] Department of Radiation Oncology, Icahn School of Medicine at Mount Sinai, New York, NY, USA. [9] Department of Biomedical Engineering, Columbia University, New York City, NY, USA. [10] Vagelos College of Physicians & Surgeons, Columbia University, New York City, NY, USA. [11] Department of Radiation Oncology, University Hospitals Seidman Cancer Center, Case Western Reserve School of Medicine, Cleveland, OH, USA. ✉email: nicholaszaorsky@gmail.com

Cancer is the second leading cause of death in the United States (US); in 2018 there were over 600,000 deaths attributable to cancer[1]. While it is commonly reported over 90% of cancer deaths are from metastatic cancer, an exact figure is unknown[2]. In 2018, approximately 280,000 of the 600,000 cancer deaths were made up of just the 6 most common primary metastatic cancers[3]. A prior projection indicates that the number of living cancer survivors will increase by 68%, and be characterized by an expanding geriatric population by 2040[4]. Taken together, these projected trends will be accompanied by a redistribution in the epidemiology of metastatic disease[4]. Studies have classically focused on the epidemiology and survivorship of local and locally advanced disease for specific cancer subtypes[5–7]. Thus, there is an unmet need to explore long-term survivorship of metastatic cancer in the US.

Recent data has demonstrated that certain subgroups of patients with metastatic cancer are living longer than in the past[8–10]. The introduction of metastases directed therapy has improved survival outcomes in patients with low metastatic burden across certain cancer subtypes[11]. This led to the develop a novel staging system for patients living with metastatic cancer which advances' prognostication by accounting for site, tumor, age, race, and sex (STARS). The STARS system allows for fine-tuned subclassification of metastatic disease into Stages IVA-E[12]. STARS IVA and IVB patients groups have been projected to contain a greater proportion of long-term survivors, as these patients typically have indolent cancers such as with prostate cancers[12]. Given that metastatic disease outcomes are evolving, projecting future incidence to continue improvements in targeted prevention and intervention, health policy and clinical intervention efforts, and allocations of healthcare resources is of great interest.

Our study aims to address this gap in knowledge by exploring the projected distribution and long-term (5-year) survivorship of metastatic cancer. The objectives of this work are to (1) report the epidemiology of metastatic cancer in the US with respect to past, current, and forecasted trends in incidence and (2) estimate odds of long-term survivorship until 2040. We found from 2018 to 2040 the APC in incidence of metastatic cancer in the US will decrease by 0.70 per 100,000 individuals, and long-term survivorship will increase by 46.7%. The results of this study may be useful in understanding the epidemiological trends in metastatic cancer and potentially influence policy and health care delivery efforts that are tailored for patients with metastatic cancer.

## Methods

**Data acquisition and analysis**. In this population-based study, patients with metastatic cancer, diagnosed between 1988 and 2018, were abstracted from the National Cancer Institute's Surveillance, Epidemiology, and End Results (SEER) database[13]. SEER is a network of population-based tumor registries from geographically distinct regions in the US, chosen to represent the racial and ethnic heterogeneity of the country. For this study, the SEER 9 database was used, which represents 9% of the US population and encompasses data from 1975 to 2018. The SEER registry includes data on incidence, survival, treatment (limited and upon request), sex, age at diagnosis, race, marital status, and year of diagnosis. SEER*Stat 8.3.9 was used for this analysis. Patients diagnosed only through autopsy or death certificate were excluded. All incidence rates were age-adjusted to the 2000 US standard population and reported per 100,000 individuals. Overview, registry selection, limitations, instructions for access of SEER and data availability of SEER software's are described in Supplemental Methods. SEER data is deidentified and attested to through a formal determination by a qualified expert as defined in

Section §164.514(b)(1) of the HIPAA Privacy Rule[14]. This formal determination by a qualified expert waives necessity of Western Institutional Review Board (IRB) review. All data used to reproduce the figures and tables in this work can be accessed in the Supplemental Data 1–9. Additional analyses were conducted using Microsoft Excel 16.16.10 (Microsoft, Redmond, WA), MATLAB version R2021a and R2021b (MathWorks Inc., Natick, MA) and R (R foundation for Statistical Computing, Vienna, Austria). Patients were classified by cancer type and year based upon their first metastatic diagnosis.

**Incidence projection and trend models**. Descriptive quantification data of patients with metastatic disease was obtained from SEER. US population at risk values for future timepoints were extrapolated from the linear regression model fitted to the past population at risk values[15].

*Age–period–cohort forecasting models*. We utilized the method developed by Yang et al. to predict future cancer incidence/mortality using age-period-cohort models. For detailed method description, please refer to the article and book[15].

As an example, we summarized the steps used to calculate expected future incidence rate.

1) We rearranged the input data so that the age and period intervals are equal (i.e. 1-year age groups and 1-year calendar periods).

2) We estimated age-period-cohort model via intrinsic estimator (IE)[16].

3) We projected APC model coefficients using a autoregressive integrated moving average (ARIMA) model[15].

4) We constructed 95% prediction intervals according to the percentile bootstrap interval with 1,000 bootstrap samples (i.e., interval between the 25th quantile value and the 975th quantile values of the bootstrap parameter estimates)[17].

5) Lastly, we calculated the expected incidence rates across age groups to get the total rate at each future time point. It should be noted that the population at risk values for future timepoints were extrapolated from the linear regression model fitted to the past population at risk values (1988–2018; 1-year interval) obtained from SEER.

*AAPC calculation from 1988 to 2040*. The ARIMA APC model was used to predict incidence with a 95% confidence interval of individual metastatic diseases, and metastatic disease to specific sites, among ages 0 to 85 from 2019 to 2040. We analyzed 13 most prevalent cancer types using the data abstracted from SEER 9 (for cancer incidence and long-term survival odds). For each cancer type, incidence by age from 0–84 and single calendar year (from 1988 to 2018) were directly obtained from SEER. For each calendar year, we used age-adjusted incidence rates to US 2000 standard population across all age groups to calculate a total rate for all age groups by weighted averaging in which the weight is the size of the population. The AAPC and its confidence interval was then directly calculated from total rates by using Joinpoint Regression Program, which calculated a weighted average of the APC. For reproducibility, all settings used in the Joinpoint Regression Program can be found in the template file (.jpt) provided in supplemental data file "Joinpoint Template".

*Joinpoint analysis and jump model*. Joinpoint regression models were employed to analyze long-term trends in metastatic cancer incidence from the original SEER data and the ARIMA predictions (1988–2040). The joinpoint model is a well-established methodology for modeling trends over time using connected linear segments, usually on a logarithmic scale. Joinpoint software

determines changes in trends by connecting piecewise linear fits on a logarithmic scale at various "joinpoints[18]." The software started from a value of 0 joinpoints and increased the number to a maximum of 3 to determine the appropriate number of join-points that allows for the best statistical fit using a Monte Carlo permutation method[19]. Average annual percent change (AAPC) was used to determine if any significant changes in overall trends from 1988–2018 were present. Annual percent change (APC) was calculated for each line segment and 95% confidence intervals were estimated. For each line segment, APCs were tested to determine if any differences existed from a value of 0 (null hypothesis). The null hypothesis was rejected for $p < 0.05$; all statistical tests were two-sided. APCs describe changes in trends at each joinpoint, and the final trend forecasting each cancer subtype was analyzed in the final output[20]. Newer staging system are added by cancer registries to keep definitions consistent with the current understanding of diseases. Such coding changes may cause discontinuous increases/decreases, or "jumps," in the data series, even though it may not affect the underlying trend[21]. Joinpoint models that ignore data jumps due to coding changes may produce biased estimates of trends. Details of the assumptions, equations, and uses of joinpoint jump model can be found here[21].

**Survival projection.** Long-term survivorship was defined as a binary outcome with patients classified as alive or dead at follow-up at least 60 months after primary diagnosis. Living patients with follow-up time less than 60 months were excluded from the analysis because it was not possible to determine whether or not they would eventually be classified as long-term survivors. Thus, the patient population included (1) deceased patients with survival times less than 60 months and (2) all patients with follow-up times greater than or equal to 60 months. For each year from 1988 to 2013, odds ratios (ORs) of long-term survival vs. 1988 were computed using the epitools R package[22]. The last year included in this analysis was 2013 as data from 2014 to 2018 was excluded due to the inability to determine long-term survival (>60 months). A strong linear trend was observed when the ORs were plotted by year. A simple linear regression model was fit, and this model was used to extrapolate ORs of long-term survivorship from 2014 to 2040. Joinpoint software's "Jump Model" was applied for fitting, to account for the addition of databases and staging changes within SEER, which caused a "jump" in data beginning in 2004. Additional details regarding our methodology to analyze survival projections are in the supplemental methods.

**Reporting summary.** Further information on research design is available in the Nature Portfolio Reporting Summary linked to this article.

## Results
**Cancer incidence trends.** During 2015, there were a total of 20,798 patients with the index metastatic cancers studied in National Cancer Institute's Surveillance, Epidemiology, and End Results (SEER) database sample population. The incidence of metastatic disease was 53 per 100,000 individuals. The age–period–cohort autoregressive integrated moving average (ARIMA) model predicted that by 2040, there would be 12,285 patients living with metastatic cancer leading to an incidence of 34 per 100,000 individuals. A full breakdown of metastatic cancers incidence and count of patients can be found in Supplemental Table 1; the distribution of metastatic cancer incidence by age can be found in Supplemental Fig. 1.

The three most common metastatic primary sites include the lung and bronchus, colon and rectum, and pancreas (Fig. 1a).

Figure 1b presents patient counts of ARIMA fitted projections from 1988 to 2040. In 2015, metastatic cancers of the lung and bronchus, colon and rectum, and pancreas had incidences of 27.8, 8.48, and 6.85 cases per 100,000 individuals, respectively. By 2040, the predicted incidences of lung and bronchus and colon and rectum decreased to 7.80 and 4.17 per 100,000 individuals, respectively, and the incidence of pancreatic cancer increased to 7.20 per 100,000 individuals.

Figure 2 shows the incidence of metastases to prevalent locations; lung, bone, liver, and brain. Our analyses predict a decrease in metastases to lung ((APC (2019–2030) = −1.90, 95% CI [−2.90, −1.00]; $p < 0.001$), (2030–2040) = −3.70, 95% CI [−4.60, −2.80]; $p < 0.001$)), liver (APC = −3.40, 95% CI [−3.50, −3.30]; $p < 0.001$), bone (APC = −4.00, 95% CI [−4.30, −3.70]; $p < 0.001$), and brain (APC = −2.30, 95% CI [−2.60, −2.00]; $p < 0.001$) to 2040. Lung cancers were the most common primary tumor to metastasize to the lung, bones, or brain. Pancreatic, colorectal, and lung cancers were the most common primary tumor to metastasize to the liver. Jointpoint analysis of trends in incidence of metastases to the lung, bone, liver, and brain are provided in Supplemental Fig. 2.

Figure 3 displays ARIMA fitted and predicted models of metastatic cancer incidence to 2040. Supplemental Fig. 3 provides joinpoint analysis of metastatic incidence trends. From 1988 to 2018, there were statistically significant increases in AAPC of pancreatic, kidney, melanoma, and liver cancer sites, significant decreases in AAPC of colon and rectal, ovary, and prostate cancers and no significant change in esophagus, breast, corpus uteri, lung and bronchus, bladder, and stomach cancer AAPC (Table 1).

Overall incidence of metastatic cancer is predicted to decrease at a rate of 0.70 cases per 100,000 individuals per year (APC = −0.70, 95% CI [−1.10, −0.40]; $p < 0.001$). Metastatic cancers predicted to increase in incidence are esophageal (APC = 0.45, 95% CI [0.04, 0.87]; $p = 0.032$), pancreatic (APC = 1.66, 95% CI [1.40, 1.90]; $p < 0.001$), kidney (APC = 0.40, 95% CI [0.10 0.60]; $p < 0.001$), melanoma of the skin (APC = 2.53, 95% CI [2.40, 2.70]; $p < 0.001$), and breast (APC = 1.84, 95% CI [1.24 2.45]; $p < 0.001$). Metastatic cancer of the lung (APC = −3.44, 95% CI [−3.73, −3.16]; $p < 0.001$), liver and intrahepatic bile duct (APC = −1.14, 95% CI [−2.10, −0.18]; $p = 0.021$), ovaries (APC = −1.62, 95% CI [−2.05, −1.18]; $p < 0.001$), and prostate (APC = −1.28, 95% CI [−2.22, −0.34]; $p = 0.009$) are predicted to decrease in incidence. Metastatic cancers not projected to change significantly in incidence include colon and rectal (APC = −0.41, 95% CI [−0.89, 0.07]; $p = 0.089$), corpus uteri (APC = −0.40, 95% CI [−0.86, −0.07]; $p = 0.091$), bladder (APC = −1.02, 95% CI [−2.13, 0.11]; $p = 0.076$), and stomach (APC = −0.61, 95% CI [−1.27, 0.06]; $p = 0.073$). When these data are stratified by age, the incidence of each metastatic cancer is much greater for patients above age 60 (Supplemental Fig. 1). Metastatic cancers of the colon, prostate, bladder, and skin (melanoma's) were particularly correlated to advanced age. Heatmaps provided illustrate increasing incidence was correlated more with age than primary site.

**Changes in long term survivorship.** From 1988–2013, there was a strong increasing linear trend in long-term survivorship of metastatic disease overall. Long term survivorship prediction and odds ratio for all metastatic subtypes is shown in Fig. 4a and Supplemental Table 2. Metastatic cancers with the greatest incidence were those of lung and bronchus, colon and rectum, and breast. The odds ratio of long-term survivorship by 2013 increased to 3.43, 2.62, and 2.04 for these cancers, respectively. We found patients diagnosed with metastatic cancer in 2040 were

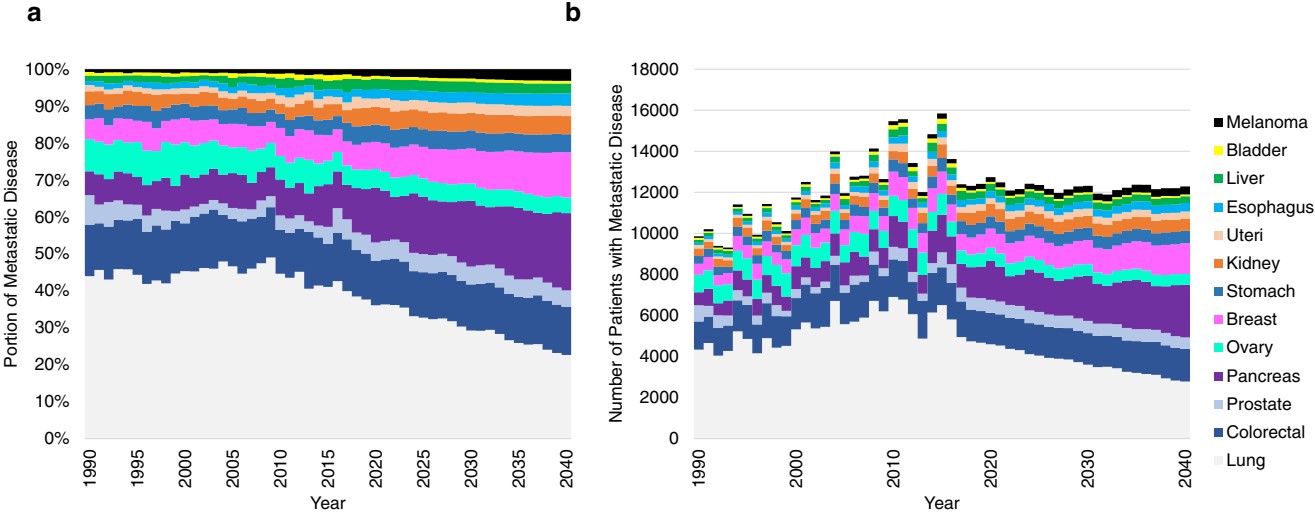

**Fig. 1 Population of patients living with metastatic cancer. a** The *y*-axis depicts the relative contribution of each metastatic cancer subtype to the total pool of metastatic disease. The *x*-axis represents the year of diagnosis. Each color depicts a disease primary site and indicates the portion of metastatic cancer made up by a given disease. **b** The *y*-axis represents the number of unique cases of metastatic cancer at diagnosis. The *x*-axis depicts the year of diagnosis. The colors depict the disease primary sites and total number of metastatic patients.

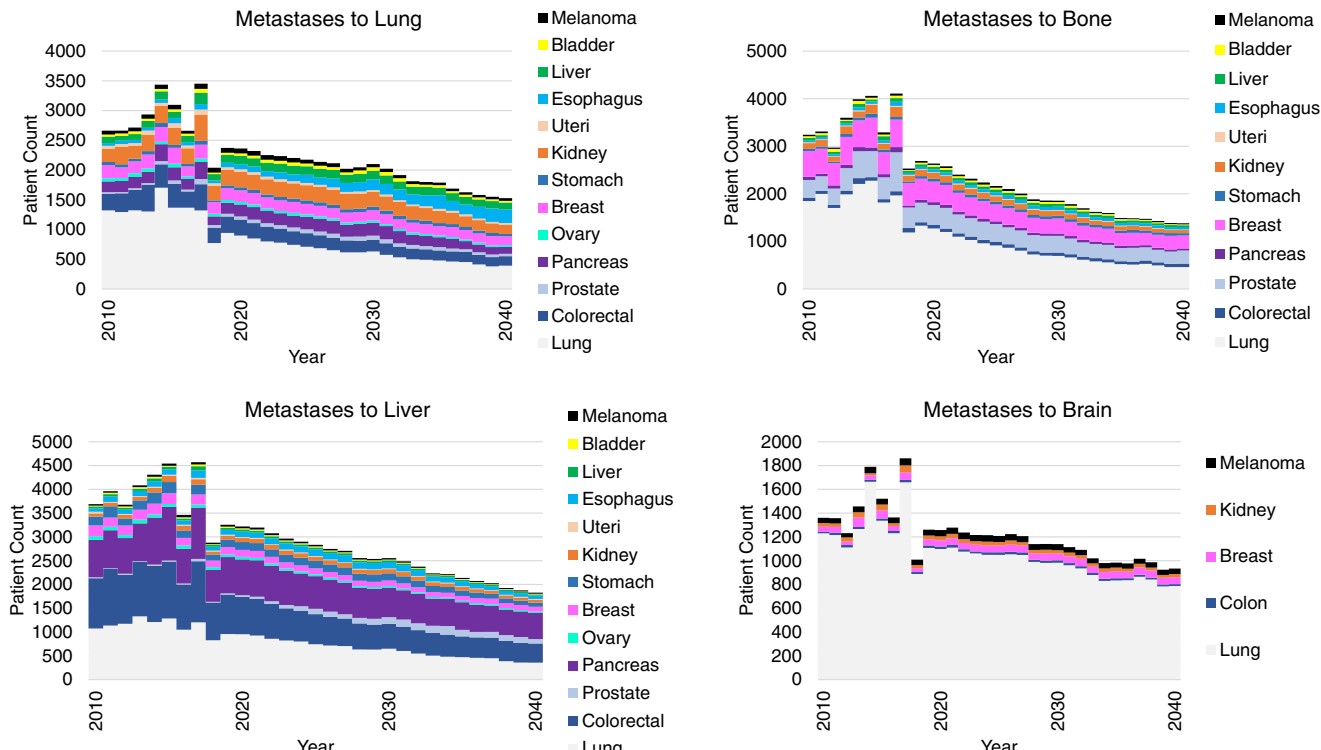

**Fig. 2 Absolute number of patients living with metastases by site of metastases.** The figure shows changes in absolute number of patients with metastases to the given site. The *x*-axis represents year, the *y*-axis number of patients. The individual colors represent the different primary disease sites.

predicted to have 46.7% greater odds of long-term survival compared to the patients diagnosed in 2013 (Fig. 4b).

## Discussion

This cross-sectional investigation presents a previously undescribed analysis regarding incidence and survivorship in metastatic disease. Our current study forecasts that the heterogeneity of metastatic cancer will evolve, driven by significant alterations in the incidence of lung, colorectal, and breast cancers. Our predictions indicate from 2018 to 2040 the APC in incidence of

metastases in the US will decrease by 0.70 per 100,000 individuals, and long-term survivorship will increase by 46.7%. Studies have shown the incidence of cancer in the US is increasing with the increase in the elderly population[4]. Similarly, our findings demonstrated that the greatest incidences of metastatic cancers in individuals aged over 75, suggesting these populations should be emphasized in ongoing surveillance efforts.

The evolving landscape of metastatic disease can be attributed to dietary and obesity trends, environmental factors, and improved screening practices. Increases in incidence of metastatic kidney cancer projections may be explained by the growing

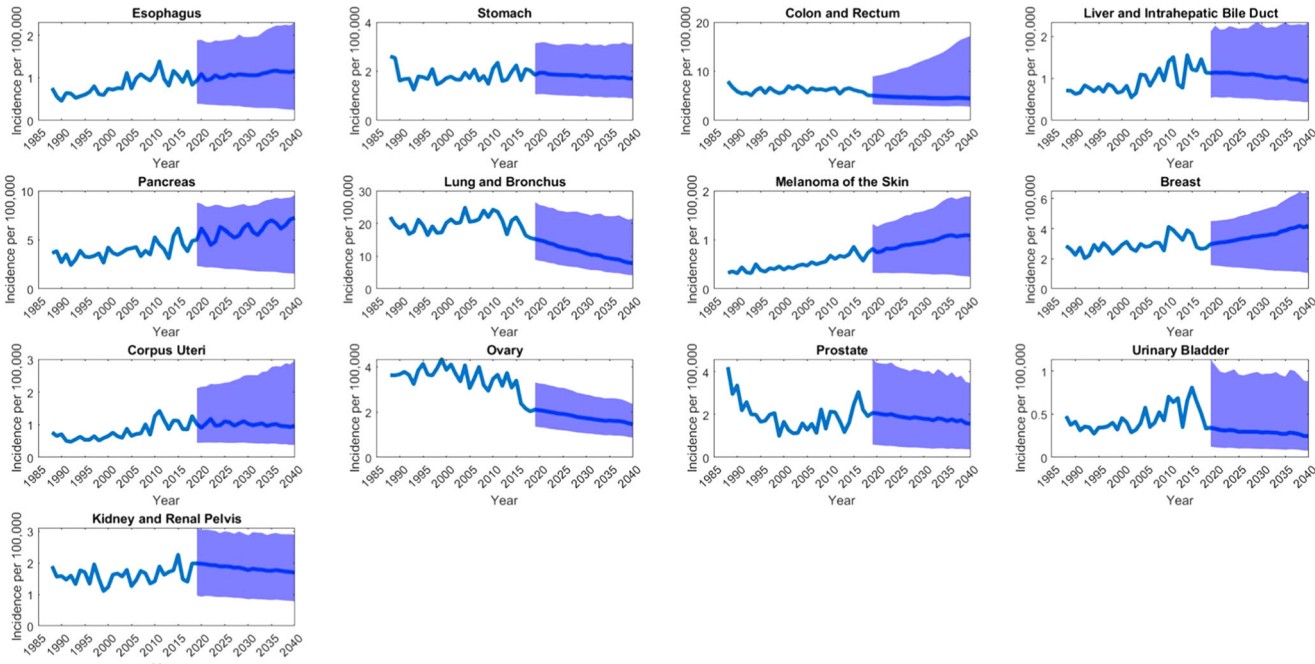

**Fig. 3 Incidence of metastatic cancer.** The figure shows population adjusted average incidence of metastatic cancers reported per 100,000 individuals. *y*-axis represents the incidence, while *x*-axis represents the year. Prediction beginning in 2019 includes confidence band represented in light blue.

**Table 1 APC and AAPC of trends in metastatic cancer**

| Cancer site | Joinpoint analysis (1988–2040) | | | | | | | | 1988–2018 |
| | Trend 1 | | Trend 2 | | Trend 3 | | Trend 4 | | |
| | Years | APC | Years | APC | Years | APC | Years | APC | AAPC |
|---|---|---|---|---|---|---|---|---|---|
| All sites | 1988–1991 | −7.30 | 1991–2015 | 0.90* | 2015–2018 | −7.10 | 2018–2040 | −0.70* | −0.80 |
| Esophagus | 1988–1990 | −16.6* | 1990–2007 | 3.90* | 2007–2040 | 0.45* | – | – | 1.13 |
| Pancreas | 1988–2040 | 1.66* | – | – | – | – | – | – | 1.66* |
| Kidney | 1988–2040 | 0.40* | – | – | – | – | – | – | 0.40* |
| Melanoma | 1988–2040 | 2.52* | – | – | – | – | – | – | 2.52* |
| Breast | 1988–2014 | 1.54* | 2014–2017 | −7.14 | 2017–2040 | 1.84* | – | – | 0.65 |
| Colorectal | 1988–1990 | −14.64 | 1999–2014 | 0.470* | 2014–2020 | −4.70* | 2020–2040 | −0.41 | −1.31* |
| Corpus uteri | 1988–1992 | −9.20 | 1992–2007 | 2.55* | 2007–2010 | 12.81* | 2010–2040 | −0.40 | 1.08 |
| Ovary | 1988–2014 | −0.42* | 2014–2017 | −14.29 | 2017–2040 | −1.62* | – | – | −1.94* |
| Lung | 1988–1992 | −5.04 | 1992–2009 | 1.63* | 2009–2040 | −3.44* | – | – | −0.82 |
| Bladder | 1988–1994 | −6.03 | 1994–2016 | 3.50* | 2016–2019 | −20.47 | 2019–2040 | −1.02 | −0.25 |
| Liver | 1988–2017 | 2.33* | 2017–2040 | −1.14* | – | – | – | – | 2.21* |
| Stomach | 1988–1990 | −24.68* | 1990–2018 | 0.66* | 2018–2040 | −0.61 | – | – | −0.80 |
| Prostate | 1988–2002 | −7.16* | 2002–2016 | 4.47* | 2016–2040 | −1.28* | – | – | −1.51* |
| **Metastases site** | Joinpoint analysis (2010–2040) | | | | | | | | 2010–2018 |
| | Trend 1 | | Trend 2 | | Trend 3 | | Trend 4 | | |
| | Years | APC | Years | APC | Years | APC | Years | APC | AAPC |
| Lung | 2010–2014 | 6.30* | 2014–2019 | −7.60* | 2019–2030 | −1.90* | 2030–2040 | −3.70* | −0.70 |
| Bone | 2010–2015 | 5.10* | 2015–2019 | −10.70* | 2019–2040 | −4.00* | – | – | −1.10 |
| Liver | 2010–2015 | 2.50* | 2015–2018 | −8.50* | 2018–2040 | −3.40* | – | – | −1.80 |
| Brain | 2010–2040 | −2.30* | – | – | – | – | – | – | −2.30* |

prevalence of obesity and advancements in imaging studies leading to incidental overdiagnosis[23,24]. Improvements in screening and diagnostic testing also account for some of these changes. Coelho et al. proposed that advancements in pathology prognosis resulted in more melanoma cancer patients being adequately diagnosed[25]. Improved screening practices increased prostate cancer incidence rates after the advancement of prostate-specific antigen tests in the 1990s. Our model's predicted decrease in prostate cancer metastases may be secondary to a decrease in primary prostate cancer diagnosis rates[5]. Namely due to US

Preventative Services Task Force grade D recommendations in May of 2012 against routine screening among men of all ages[26,27]. While it is possible that metastatic prostate incidence may stabilize or decrease with advances in imaging modalities, prostate MRI or prostate-specific membrane antigen PET (2018) could increase incidence of lower stage prostate cancer via improved prognostication[28,29].

The present work also forecasts a sharp decline in lung cancer incidence, similar to other population based studies[30–32]. Dela Cruz et al. found that the overall decrease in cigarette use in the

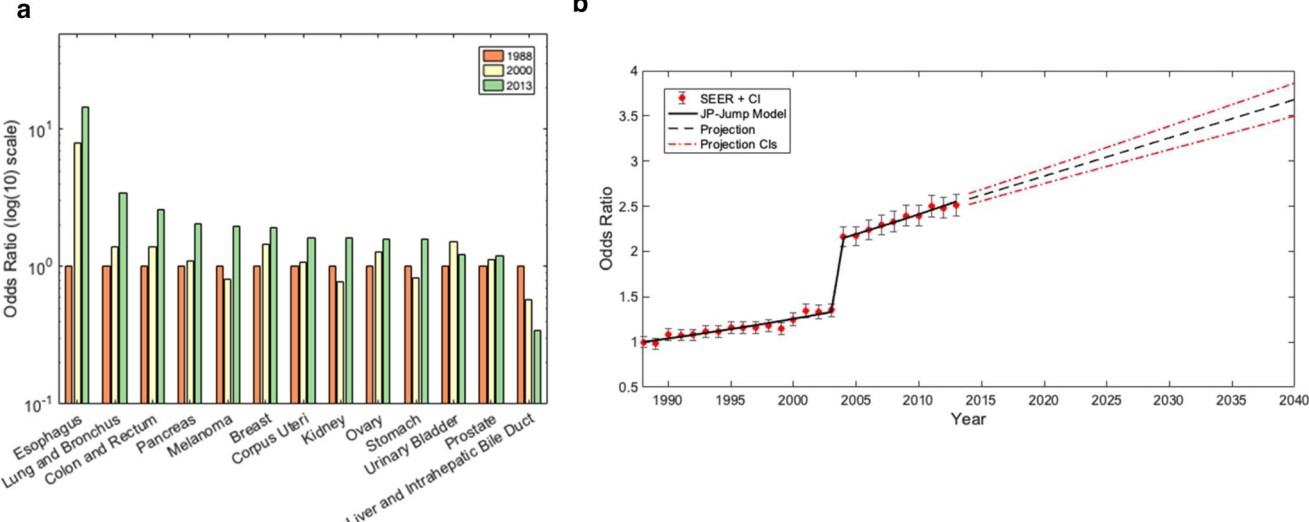

**Fig. 4 Long term survivorship odds ratios vs. 1998. a** Shows projected odds ratios (ORs) of long-term metastatic cancer survivorship using Joinpoint jump model. 1988 was used as a reference for rate of long-term survivorship ($\geq 60$ months). Y-axis represents ORs while x-axis represents year. The dashed black line represents projected odds ratio while the dashed red represents the confidence interval. **b** The y-axis depicts the odds ratio of long-term survivorship of individual metastatic cancers currently registered in SEER. Different time periods (1988 vs 2000 vs 2013) are shown in red, yellow, and green, respectively.

US may contribute to these trends[32]. In past studies of synchronous metastases, lung primary tumors were the top contributor to metastases of the brain, bones, and lungs[33–35]. Thus, the forecasted decreases in metastatic lung cancer incidence may be the primary driving force behind the predicted decreases in the number of metastases to liver, lung, bone, and brain.

We illustrate that long-term (5-year) survivorship has improved in most metastatic cancer subtypes, likely due to advances in systemic therapy and is projected to improve in the future. For example, immune checkpoint inhibitors, targeted agents, and the incorporation of ablative radiotherapy into the management of oligometastatic cancer have all improved patient outcomes[11,36,37]. Notably, esophageal cancer had the greatest increase in odds of long-term survival in this study. Njei et al. found that from the 1970s to 2000s, the incidence of esophageal adenocarcinoma doubled, while the incidence of squamous cell carcinoma decreased[38]. This change in heterogeneity, earlier detection, and utilization of treatment modalities, such as neoadjuvant chemo-radiotherapy or surgery, may explain this improvement in long-term survival[38,39].

Our findings have several clinical implications. The decreasing incidence of metastases of the lung and liver combined with increasing incident rates of breast and melanoma cancers may shift the pool of metastatic disease to tumors more heavily screened for and monitored, which in turn leads to earlier intervention and better overall survival. The decreasing incidence rates paired with increased survival can be explained by the Will Rogers phenomenon; whereby diagnostic improvements may lead to this earlier detection and improved prognosis[40]. As a result of favorable stage migration, long-term survival is improving in metastatic disease even in cases where advances in treatment modalities have not occurred.

Our forecasting is limited by rapid development of treatment and technologies that may improve cancer screening, prevention, or treatment. Additionally, due to sudden changes in SEER staging classifications, the SEER mortality data had large "jumps" in values, which limited our use of a survivorship-period-cohort model, which may have led to more accurate predictions. With SEER data, our analyses are restricted to incident metastatic cancers and are not able to assess for cancers that may have been localized at diagnosis but then progressed or recurred as metastatic. As such, our estimates likely under-represent the number of patients diagnosed and living with metastatic cancers. Further limitations of SEER database specifically are available in supplemental methods.

## Conclusion

In this cross-sectional, US population-based study, metastatic cancer was shown to have unique epidemiological patterns with improved long-term survival. By 2040, the distribution of patients living with metastatic cancer is predicted to shift in predominance from metastatic cancers traditionally diagnosed late to those heavily screened for and monitored. Continued research on metastatic cancers is important to understanding and addressing the distinct medical needs of this patient population.

## Data availability

Instructions for accessing SEER datasets are detailed in the supplementary methods file. All the data files supporting the findings of this study are available within supplementary data files. Specific SEER inputs to replicate this data extraction can be found in Supplemental Data 1 "SEER Output," and Supplemental Data 3 "SEER Mets to Brain, Bone, Liver, Lungs," under the individual "info" tabs. Supplemental Data 2 "ARIMA Model Output" contains data extrapolated to create Figs. 1 and 3, Supplemental Figs. 1 and 3, Table 1 (top), and Supplemental Table 1. Supplemental Data 5–8 "Mets to …." contain data extrapolated to create Fig. 2, Supplemental Fig. 2, and Table 1 (bottom). Supplemental Data 9 contain the SEER case listing file used to produce Fig. 4. All data can also be accessed and retrieved publicly from the NCI website, as outlined in steps 1–3 above.

## Code availability

The Joinpoint template used in this paper can be accessed in Supplemental Data 3.

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

## Acknowledgements

We thank Menglu Liang of the Wang lab in Penn State College of Medicine department of bioinformatics and genomics for her guidance with the Joinpoint program. We thank Dr. Saad Sheikh of University Hospitals for his assistance in editing of this manuscript.

## Author contributions

The authors confirm contribution to the paper as follows: study conception and design: N.G.Z.; data collection: N.L.H., K.A.M., N.G.Z.; analysis and interpretation of results: N.L.H., K.A.M., C.K., V.W., M.W., E.J.L., M.R.H., D.J.L., N.G.Z.; draft manuscript preparation: N.L.H., K.A.M., C.K., E.J.L., M.R.H., L.N., D.E.S., N.G.Z. All authors reviewed the results and approved the final version of the manuscript.

## Competing interests

The authors declare no competing interests.
