## [Peer Review File · Communications Medicine]

Reviewers' comments:

Reviewer #1 (Remarks to the Author):

The authors have done a good job by assessing and projecting the burden of metastatic cancer to 2040 by characterizing past, current, and forecasted incidence trends and also estimating odds

of long-term (5-year) survivorship in USA. However, the following changes are required.

1. Mention the full form of APC in abstract.

2. Line 53 to 54 states that majority of patients die of metastatic cancer. It is advised to incorporate the exact statistics of the burden (in terms of both incidence and deaths) of metastatic cancer in the USA and worldwide.

3. The full form of abbreviations such as SEER and ARIMA should also be mentioned in the main manuscript.

4. Elaborate the age-wise burden in the results section.

5. Authors should incorporate the past, present and future estimates of metastatic cancer burden by gender as well.

6. Authors have estimated the burden of metastatic cancer by incidence and survivorship. Death burden should also be mentioned across age, gender, and sites.

7. Lastly, authors are advised to thoroughly read the manuscript and remove all grammatical errors. For instance: Line 95, "Our analyses predict an increase in metastases to the liver to 2040...", is grammatically incorrect.

Reviewer #2 (Remarks to the Author):

This paper analyzed the incidence and long-term survival of metastatic cancer and predicted the future trend in the United States. Comments below:

1. The term "metastatic site" may create confusion. It may refer to the primary site from which the metastasis originates or where it resides. Suggest the authors make the term explicit.

2. The authors use the ARIMA model for analysis. But the model only considers time ("period"), but not "age" and "cohort". I suggest the authors use the "age-period-cohort model: for trend analysis and forecasting.

3. Likewise, the authors use a log-linear odds-ratio model to analyze and forecast long-term survival. I suggest the authors use the "survivorship-period-cohort model" (reference: a survivorship-period-cohort model for cancer survival: application to liver cancer in Taiwan, 1997–2016. *Am J Epidemiol* 2021;190:1961-1968) for their purpose.

4. The apparent discontinuity in the observed and predicted trends in Figures 1 and 2 indicates poor goodness of fit of the models.

Thank you for your helpful comments regarding our work titled, “Future incidence and long-term survival of metastatic cancer in the United States.” We have addressed all comments in a point-by-point response below. We provide the reviewer’s comments (in black) and our responses (in red).

Reviewers' comments:

Reviewer #1:

The authors have done a good job by assessing and projecting the burden of metastatic cancer to 2040 by characterizing past, current, and forecasted incidence trends and also estimating odds of long-term (5-year) survivorship in USA. However, the following changes are required.

1. Mention the full form of APC in abstract.

AUTHOR RESPONSE: Thank you for this helpful comment. We have mentioned the full term of APC prior to first use in abstract.

2. Line 53 to 54 states that majority of patients die of metastatic cancer. It is advised to incorporate the exact statistics of the burden (in terms of both incidence and deaths) of metastatic cancer in the USA and worldwide.

AUTHOR RESPONSE: Thank you for this valuable suggestion. We agree this is important to include and there is sparse literature on exact burden of metastatic disease. A recent report from the JNCI provided an estimate of disease burden for the US alone. Nearly half of all cancer deaths were due to only the top 6 most common primary metastatic cancers: <https://doi.org/10.1093/jnci/djac158>.

3. The full form of abbreviations such as SEER and ARIMA should also be mentioned in the main manuscript.

AUTHOR RESPONSE: Thank you for this helpful comment, we have updated to define each acronym before initial use in the main manuscript.

4. Elaborate the age-wise burden in the results section.

AUTHOR RESPONSE: Thank you for this helpful comment, we have included a line highlighting the relationship with age and cancers of the colon, prostate, bladder and melanomas. All metastatic cancer incidence increased with age, with greatest increases found over 60, these however tended to be found primarily in the 75+ population. No cancers were found to have bimodal or distributions to younger populations, therefore we did not focus significantly on age-wise burden.

5. Authors should incorporate the past, present and future estimates of metastatic cancer burden by gender as well.

AUTHOR RESPONSE: Thank you for this helpful point. We originally divided groups based upon gender; however, predictions had large confidence intervals, and we are unable to accurately comment on incidence by gender. Thus, gender was not included in this paper. However, we have another paper in submission, currently, estimating past, present, and future estimates of cancer burden, which considers gender. To replicate this analysis for metastatic cancer would be an entirely new project idea, but one we may consider in the future.

6. Authors have estimated the burden of metastatic cancer by incidence and survivorship. Death burden should also be mentioned across age, gender, and sites.

AUTHOR RESPONSE: Thank you for this helpful comment. Our concentration was the ongoing burden of metastatic disease including the increasing survivorship for the overall patient population in SEER. For this reason, we focused our results to index disease and survivorship. Death burden of metastatic cancer is a valuable topic and based on our literature search there is very little literature examining it. This can be an entirely new project.

7. Lastly, authors are advised to thoroughly read the manuscript and remove all grammatical errors. For instance: Line 95, “Our analyses predict an increase in metastases to the liver to 2040...”, is grammatically incorrect.

AUTHOR RESPONSE: Thank you for this helpful comment. We have amended this line, as well as updated order of supplemental figures as these were found to be misnumbered.

Reviewer #2:

This paper analyzed the incidence and long-term survival of metastatic cancer and predicted the future trend in the United States. Comments below:

1. The term “metastatic site” may create confusion. It may refer to the primary site from which the metastasis originates or where it resides. Suggest the authors make the term explicit.

AUTHOR RESPONSE: Thank you for this helpful suggestion, we agree this can create significant confusion. We updated appropriate lines to specify primary and secondary disease sites to provide further clarification for readers.

2. The authors use the ARIMA model for analysis. But the model only considers time (“period”), but not “age” and “cohort”. I suggest the authors use the “age-period-cohort model: for trend analysis and forecasting.

AUTHOR RESPONSE: Thank you for this comment. We did use an age-period-cohort model in this analysis, but in the methods section only mentioned the ARIMA aspect. In fact, we use an age-period-cohort autoregressive integrated moving average model, and we have updated the methods to make this clear.

In our supplementary materials of our original manuscript submission, we provided a detailed explanation of our age-period-cohort forecasting models. However, we failed to direct readers and the reviewer to this valuable further explanation of our methods, so we thank the reviewer for this great catch.

To address this, we include the following sentence in our methods of the paper in case readers would like to understand the methods in more depth: “A more detailed explanation of our modeling approach can be found in the supplementary materials.”

Last, we have included an excerpt of the explanation from our initial manuscript submission for the reviewer, below. In summary, we used an APC method developed by Yang et al. We rearranged input data so that the age and period intervals were equal, estimated the APC model via intrinsic estimator, projected APC model coefficients using ARIMA, constructed 95% CI according to the percentile bootstrap interval with 1,000 bootstrap samples (i.e., interval

between the 25th quantile and the 975th quantile values of the bootstrap parameter estimate), and calculated the expected incidence rates across age groups to output the total rate at each future time point.

“AAPC calculation from 1988 to 2040

We analyzed 13 most prevalent cancer types using the data abstracted from SEER 9 (for cancer incidence and long-term survival odds). For each cancer type, incidence by age from 0-84 and single calendar year (from 1988 to 2018) were directly obtained from SEER. For each calendar year, we used age-adjusted incidence rates to US 2000 standard population across all age groups to calculate a total rate for all age groups by weighted averaging in which the weight is the size of the population. AAPC and its confidence interval is then directly calculated from total rates by using Joinpoint Regression Program, which calculated a weighted average of the APC. For reproducibility, all settings used in the Joinpoint Regression Program can be found in the template file (.jpt) provided on GitHub.

Age-Period-Cohort forecasting models

We utilized the method developed by Yang et al. to predict future cancer incidence/mortality using age-period-cohort models. For detailed method description, please refer to the article and book^{3,4}.

As an example, we summarized the steps used to calculate expected future incidence rate.

- 1) We rearranged the input data so that the age and period intervals are equal (i.e. 1-year age groups and 1-year calendar periods).
- 2) We estimated age-period-cohort model via intrinsic estimator (IE)⁴.
- 3) We projected APC model coefficients using autoregressive integrated moving average (ARIMA) model³.
- 4) We constructed 95% prediction intervals according to the percentile bootstrap interval with 1,000 bootstrap samples (i.e., interval between the 25th quantile value and the 95th quantile values of the bootstrap parameter estimates)⁵.
- 5) Lastly, we calculated the expected incidence rates across age groups to get the total rate at each future time point.

It should be noted that the population at risk values for future timepoints were extrapolated from the linear regression model fitted to the past population at risk values (1988-2018; 1-year interval) obtained from SEER.

Joinpoint analysis and jump model

The joinpoint model is a well-established methodology for modeling trends over time using connected linear segments, usually on a logarithmic scale. Newer staging system are added by cancer registries to keep definitions consistent with the current understanding of diseases. Such coding changes may cause discontinuous increases/ decreases, or “jumps,” in the data series, even though it may not affect the underlying trend⁶. Joinpoint models that ignore data jumps due to

coding changes may produce biased estimates of trends. Details of the assumptions, equations, and uses of joinpoint jump model can be found here⁶.”

3. Likewise, the authors use a log-linear odds-ratio model to analyze and forecast long-term survival. I suggest the authors use the “survivorship-period-cohort model” (reference: a survivorship-period-cohort model for cancer survival: application to liver cancer in Taiwan, 1997–2016. *Am J Epidemiol* 2021;190:1961-1968) for their purpose.

AUTHOR RESPONSE: Thank you for this helpful comment. We use an age-period-cohort model for the incidence trends, as we wanted to explore the projected distribution in detail. However, we chose not to use an age-period-cohort model for survival for several reasons. The primary reason is due to limitations with SEER stage re-classifications that cause a pseudo-Will Rogers phenomenon in mortality. This can be seen with the large jump in data at approximately 2004, for which we provide justification in the discussions section, and cite the SEER webpage directly. SEER recommends researchers apply a SEER jump-model, tangentially related to the JoinPoint regression software we used. The log-linear odds-ratio jump model provided an easy way to address the jump in survival. To be consistent with our use SEER’s JoinPoint model, we used the SEER jump-model in this analysis and applied this to our primary output variable for objective (2) the odds ratio for long-term (5-year) survivorship. Last, given that objective (2) was not to characterize or project future mortality distribution (which would be a lofty goal, and paper in it of itself), but instead to only estimate odds of long-term (5-year) survivorship until 2040, we believe our methods are reasonable.

4. The apparent discontinuity in the observed and predicted trends in Figures 1 and 2 indicates poor goodness of fit of the models.

AUTHOR RESPONSE: Thank you for this helpful comment. We use a very commonly utilized method of APC modeling outlined by Yang et al in the following book, which is cited over 600 times:

Yang Y, Land KC. *Age-Period-Cohort Analysis: New Models, Methods, and Empirical Applications*. Taylor & Francis; 2013. doi:10.1201/b13902

And we applied the intrinsic estimator detailed in this journal article by Yang et al., cited over 450 times:

4. Yang Y, Schulhofer-Wohl S, Fu WJ, Land KC. The Intrinsic Estimator for Age-Period-Cohort Analysis: What It Is and How to Use It. *Am J Sociol*. 2008;113(6):1697-1736. doi:10.1086/587154

Both the book and the article further describe the algebraic, geometrical, and verbal applications of the APC and IE methodology, and provide model validation evidence for both the APC and IE from empirical examples and simulation exercises. We provide a further detailed summary in the supplemental information and cite both these articles, so that readers may understand the methodology better.

However, our paper is not a methods paper, and instead is an application of these commonly used statistical models to the SEER cancer data to provide broad and generalized predictions to aid cancer surveillance efforts. We do provide 95% CI for our estimates and predictions, using the percentile bootstrap interval with 1,000 bootstrap samples (i.e., interval between the 25th quantile and the 975th quantile values of the bootstrap parameter estimate), and mention in the discussion that a major limitation is that our forecasts are based on current data

and cannot account for the rapid development of treatment and technologies that may improve cancer screening, prevention or treatment.

Last, to address the reviewers concern, we introduce another limitation: “Additionally, due to sudden changes in SEER staging classifications, the SEER mortality data had large “jumps” in values, which limited our use of a survivorship-period-cohort model, which may have led to more accurate predictions.”

REVIEWERS' COMMENTS:

Reviewer #1 (Remarks to the Author):

All the comments and suggestions are addressed.

Reviewer #2 (Remarks to the Author):

The paper has been revised, taking into account my previous comments.